# Post-intervention epidemiology of STH in Bangladesh: Data to sustain the gains

**Sanjaya Dhakal[1☯], Mohammad Jahirul Karim[2☯], Abdullah Al Kawsar[2‡],
Jasmine Irish[1‡], Mujibur Rahman[2], Cara Tupps[1], Ashraful Kabir[1‡], Rubina Imtiaz[1☯]***

**1** Children Without Worms, The Task Force for Global Health, Atlanta, Georgia, United States of America,
**2** Communicable Disease Control, Ministry of Health & Family Welfare, Dhaka, Bangladesh

☯ These authors contributed equally to this work.
‡ AAK, JI and AK also contributed equally to this work.
* rimtiaz@taskforce.org

**Data Availability Statement:** All relevant data are within the manuscript.

**Funding:** These surveys were supported with generous funding from Johnson & Johnson

## Abstract

In 2008, Bangladesh initiated Preventive Chemotherapy (PCT) for school-age children (SAC) through bi-annual school-based mass drug administration (MDA) to control Soil-Transmitted Helminth (STH) infections. In 2016, the Ministry of Health and Family Welfare's Program on Lymphatic Filariasis Elimination and STH (ELFSTH) initiated district-level community impact assessments with Children Without Worms (CWW) using standardized, population-based sampling to measure the post-intervention STH burden across all ages ($\geq 1$ yr) for the three STH species. The Integrated Community-based Survey for Program Monitoring (ICSPM) was developed by CWW and was used to survey 12 districts in Bangladesh from 2017–2020. We excluded the first two district data as piloting caused some sampling errors and combined the individual demographic and parasite-specific characteristics from the subsequent 10 districts, linking them with the laboratory data for collective analysis. Our analysis identified district-specific epidemiologic findings, important for program decisions. Of the 17,874 enrolled individuals, our results are based on 10,824 (61.0%) stool samples. Overall, the prevalence of any STH species was substantially reduced to 14% from 79.8% in 2005. The impact was similar across all ages. STH prevalence was 14% in 10 districts collectively, but remained high in four districts, despite their high reported PCT coverage in previous years. Among all, Bhola district was unique because it was the only district with high *T. trichuris* prevalence. Bangladesh successfully lowered STH prevalence across all ages despite targeting SAC only. Data from the survey indicate a significant number of adults and pre-school age children (PSAC) were self-deworming with purchased pills. This may account for the flat impact curve across all ages. Overall prevalence varied across surveyed districts, with persistent high transmission in the northeastern districts and a district in the central flood zone, indicating possible service and ecological factors. Discrepancies in the impact between districts highlight the need for district-level data to evaluate program implementation after consistent high PCT coverage.

(https://www.jnj.com/), Glaxo Smith Kline (https://www.gsk.com/en-gb/home/), and Nutrition International through funding from the Government of Canada. Technical support and field implementation was supported by CWW and the Bangladesh National ELFSTH. The funding is not specific to these surveys and funders had no role in study design, data collection and analysis, decision to publish, or preparation of the manuscript.

**Competing interests:** The authors have declared that no competing interests exist.

## Author summary

Bangladesh government conducted school-based mass drug administration (MDA) for over 10 years to control soil-transmitted helminth (STH) infections. School-based evaluations of MDA indicate a reduction in STH burden among school-aged children (SAC). To further assess the impact on the community, Children Without Worms and the Ministry of Health and Family Welfare's Program on Lymphatic Filariasis Elimination and STH (ELFSTH) initiated district-level community impact surveys in 12 districts. We share the results from the latter 10 districts here (the first two pilots were excluded because of possible sampling errors).

Our analysis of 10,824 interviews and stool samples from 10 districts showed an estimated 14% of community members infected with at least one species of STH. This finding is substantially lower than the baseline STH prevalence (79.8%) estimated in 2005. Bangladesh's successful impact was achieved across all ages despite only treating SAC. Deworming source data showed significant numbers of adults and pre-school age children (PSAC) self-dewormed with locally purchased pills. Prevalence varied across the surveyed districts, with persistent high transmission in the northeastern districts and a district in the central flood zone, indicating possible ecological and service factors contributing to persistent infections. Variable impact across districts highlights the need for sub-national level data to evaluate program performance following the consistent high intervention and could be attributable to many additional factors.

## Introduction

In 2001, the World Health Organization (WHO) recommended that member states control Soil-Transmitted Helminthiasis (STH) morbidity through preventive chemotherapy (PCT) in endemic regions. The recommended guidance utilizes a school-based platform to target one high-risk group, school-age children (SAC) through mass drug administration (MDA) to achieve at least 75% coverage consistently for five years. Once this is achieved, an impact assessment survey is recommended [1]. Like many developing countries, Bangladesh bears a high burden of STH. An estimated national STH prevalence of 79.8% (44% of which was moderate-to-high intensity infection, MHII, of *A. ascaris*) among school-aged Bangladeshi children was reported in 2005 [2]. By January 2020, Bangladesh had completed 23 rounds of school-based bi-annual MDA with Mebendazole. Bangladesh receives the largest Mebendazole donation of all endemic countries (approximately 20% of the global donation) and has an excellent supply chain record over the past 5 years (SCF-NTD data: CWW retrieved 15 June 2020).

Annual coverage data from Bangladesh indicates consistent coverage of greater than 75% for more than five years before the Integrated Community-based Survey for Program Monitoring (ICSPM) surveys began in 2017 [3]. Previously, PCT coverage data was used as a proxy to indirectly evaluate the impact of deworming on the STH burden [3,4]. A major limitation of this approach was the inability to assess the true burden of disease in the community at risk because; 1.) MDA targets only SAC (a small proportion of the at-risk population), 2.) the quality of coverage data is not tested, and 3.) targeted parasites have variable sensitivity to the single drug used for MDAs [5,6]. Additionally, PCT coverage data does not include children outside schools and adults. Available evidence indicates that these additional risk populations such as pre-school-age children (PSAC) and adults, particularly women of reproductive age (WRA) are also at risk of STH infection and share a substantial disease burden [7–9].

Therefore, to better understand the community-level program impact, the Lymphatic Filariasis Elimination and STH (ELFSTH) Program of the Bangladesh Ministry of Health &

Family Welfare (MOHFW) collaborated with Children Without Worms (CWW) to conduct community-level impact assessment surveys called Integrated Community-based Survey for Program Monitoring (ICSPM) from 2017 to 2020. The main objectives of the surveys were:

i. To estimate the statistically valid prevalence of STH infection and prevalence of moderate to high-intensity infection (MHII) in PSAC, SAC, and adults (greater than 14 years old), powered to the district level, and

ii. To evaluate potential correlates of STH infection rates including sanitation & hygiene behaviors (household level) and history and source of deworming (individual level).

In this paper, we present the results of concatenated data from surveys conducted between 2017 and 2020, focusing on parasite- and age-specific prevalence and infection intensity as well as the geographic variation of STH prevalence. This paper also presents how these results are applicable for use by the ELFSTH program towards future program actions, and how this approach can assist other similarly advanced NTD programs around the world.

## Methods

### Ethics statement

Participation in the survey was voluntary and participants provided verbal consent before the main survey. Ethical clearance was obtained through the Bangladesh Medical Research Council (BMRC), who reviewed and approved the survey protocol.

### Study design

ICSPM surveys were conducted in 12 districts, representing 7 out of 8 divisions across Bangladesh. The districts were selected by the Bangladesh ELFSTH according to programmatic priorities. The objectives of the surveys were to evaluate the impact of MDA at the community level for each parasite and each risk group, to validate deworming pill intake and pill source within six months before the survey, and to assess the effect of select WASH variables at the household level. Based on age, we defined three risk groups as follows:

- 1–4 years old: pre-school age children (PSAC)

- 5–14 years old: school-age children (SAC)

- greater than 14 years old (adults)

The district was selected by the Bangladesh ELFSTH as an ideal evaluation unit (EU), as the district is the most common administrative unit for implementation decisions where these results could be utilized. While 12 districts were surveyed in all, the first two districts (Bandarban and Nilphamari) were surveyed as a pilot to test the sampling methodology and gain field experience. As some of the sampled clusters were changed by the field teams due to local challenges, we decided to exclude these two district results leaving us with 10 district data for the current analysis. The gap between the survey and the previous deworming event was at least five months in all districts. Fig 1 shows the years of each survey for the 10 districts.

The ICSPM survey is a community-based, cross-sectional survey based on probability proportional to size sampling (PPSS). The details of the ICSPM survey methodology is available on the CWW website (http://www.childrenwithoutworms.org/). Briefly, the survey design entails a cross-sectional, mixed cluster & random systematic sample methodology and has been previously detailed [10]. The ICSPM methodology primarily relies on WHO's "Assessing the epidemiology of STH during a transmission assessment survey [11]. We targeted a sample

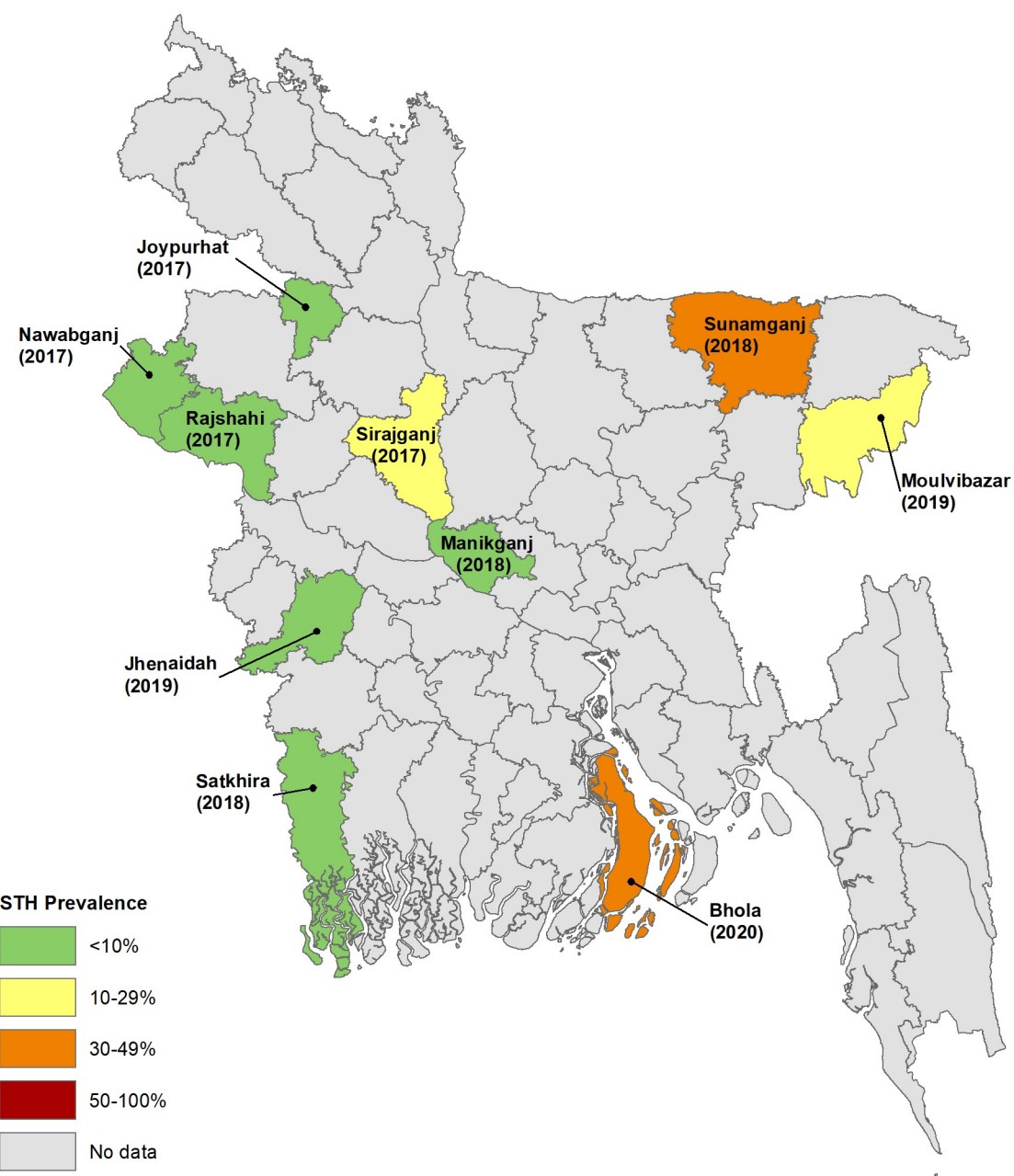

**Fig 1. Geographical distribution of surveyed districts, year of survey, and STH prevalence range.**

size of 332 for each risk group, which gave us one-sided 95% confidence for determining if the <10% prevalence, action threshold was achieved. Since the average non-response rate in the first two pilot districts was around 40%, we enrolled 465 individuals in each risk group in subsequent districts to account for this. The sampling interval was based on the proportion of each risk group within the population. The survey team used the Survey Sample Builder (SSB) tool, which was adapted to the ICSPM methodology, an excel program developed by Neglected Tropical Diseases Support Center, The Task Force for Global Health (TFGH) to select clusters and risk groups within the households.

We used the Kato-Katz method to identify and count the eggs of STH parasites following standard WHO methodology using two slides per stool specimen. Stools samples were analyzed on the same day as collected, being transported to the laboratory in a cooler box within three hours of collection. Ten percent of slides were tested blindly by another laboratory scientist for quality control. Three data sets (Household, Individual, and Laboratory) were downloaded from the secure cloud-based data-hosting platform and saved in local computers at CWW, Atlanta. After basic data cleaning, household data were first merged with individual data and later with laboratory data making one linked data file for each district. We recoded and reformatted variables as necessary to align across the districts before combining the individual data files from surveyed districts. Finally, we prepared one analytical data file for this report by stacking 10 individual data files from each of the surveyed districts. According to the 2011 national population census, the results presented here are statistically representative of about 14.1% of the Bangladeshi citizen living in those 10 districts.

We used SAS version 9.4 (SAS Inc., Cary NC, USA) to manage and analyze the data. We accounted for the cluster sample survey design in all analyses using appropriate SAS procedures. Chi-square ($\chi^2$) test was used to assess differences in prevalence between risk groups and p-values $\leq 0.05$ were considered significant. Since the survey was powered to detect the prevalence of STH and MHII down to a threshold of $\geq 10\%$ at the district level, only upper sided, 95% confidence limits are reported. We also ran some explorative analyses at the sub-district level, which lacked statistical power but provide useful insights for further program actions.

## Results

### Basic characteristics

Of 17,874 enrollees, 11,022 (61.7%) provided stool samples for laboratory examination. Subsequently, 198 (1.6%) stool sample records were excluded during the data cleaning process due to;

a.  IDs present in only one dataset

b.  duplicate IDs with mismatching data across other variables, and

c.  data entry errors.

The final "clean" dataset had 10,824 records which were used for the analysis presented here (Fig 2). The 3-most commonly reported occupation among responders were students (34.6%) housewives (25.6%), other (21.8%).

Among the 6,852 (38.3%) participants who did not provide samples, males (40.3%) were less likely to provide a stool specimen compared to their female (38.8%) counterparts (p-value 0.04). Similarly, fewer PSAC (41.1%) and SAC (40.1%) provided stool specimens compared to adults (36.8%), p-value <0.001. We believe that adults may have a more regulated bowel routine while younger kids cannot be forced to produce it at a given time.

### Prevalence and Intensity of STH Infections

The overall prevalence of any STH infection in 10 districts was 14.0% (Fig 3). There was no statistical difference in the prevalence of STH infection across the risk groups. We did not observe statistically different prevalence between females (14.4%) and males (13.4%). Of the three tested parasites, *A. lumbricoides* was the most common (10.5%) followed by *T. trichuris* (4.4%). The prevalence of hookworm was less than 1% in all risk groups. Three districts with the highest STH prevalence were Sunamganj (40.4%), Bhola (36.5%), and Sirajganj (26.9%). In

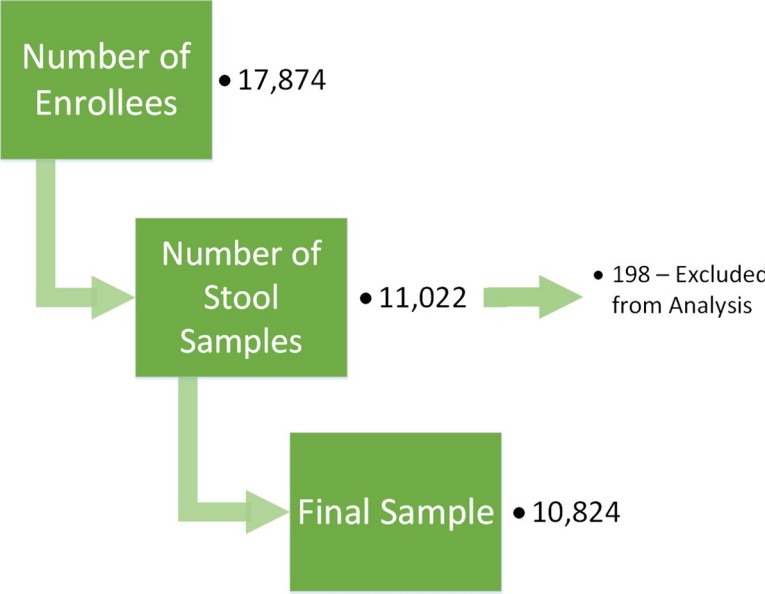

**Fig 2. Flow chart of sample selection and final number of observations.**

contrast, Satkhira (2.0%), Jhenaidah (2.4%), and Manikganj (3.1%) had the lowest STH prevalence (Fig 3).

Overall, the intensity of STH MHII in the 10 districts was 3.3%. Bhola (10.6%), Sunamganj (10.4%), Sirajganj (7.1%), and Moulvibazar (3.6%) were four districts with MHII above the WHO-recommended threshold of <1%, while the remaining 6 districts had achieved this goal with MHII ranging from 0.0% to 0.2% (Table 1). This is a significant achievement for the national program and signifies the achievement of the WHO goal of eliminating STH morbidity in majority districts.

We further explored three high prevalence (>20%) districts Sunamganj, Bhola, and Sirajganj to understand if there were any geographic concentrations of STH infection at the subdistrict level. Fig 4 illustrates the STH prevalence by sub-districts in these three high-

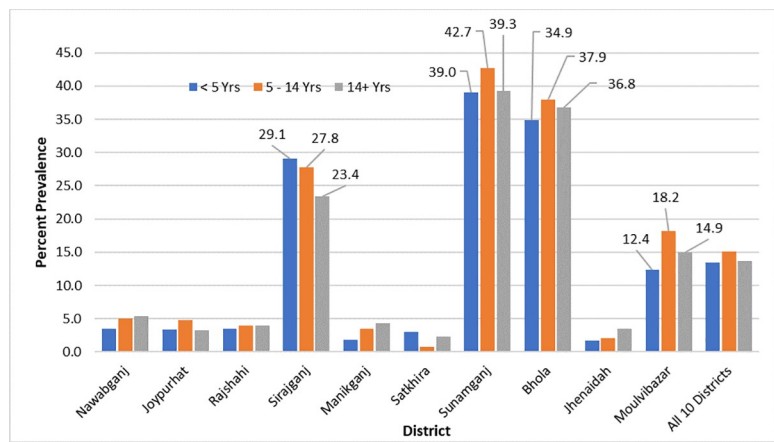

**Fig 3. Prevalence of any STH infection by district and age group.**

**Table 1. Intensity of STH morbidity by district.**

| District | Intensity of STH Morbidity among All (Prevalence %) | | | All Ages (%) |
|---|---|---|---|---|
| | **PSAC** | **SAC** | **Adults** | |
| Sirajganj | 7.0 | 6.6 | 7.8 | 7.1 |
| Sunamganj | 12.2 | 8.6 | 10.2 | 10.4 |
| Bhola | 12.0 | 11.1 | 8.5 | 10.6 |
| Moulvibazar | 3.6 | 4.8 | 2.4 | 3.6 |
| **All 10-districts** | **3.6** | **3.4** | **2.7** | **3.3** |

prevalence districts. The prevalence of STH was higher than 50% in two sub-districts (Dowara Bazar and Dakshin Sunamganj) of Sunamganj and one sub-district (Belkuchi) of Sirajganj district.

## History of deworming

The proportion of self-reported deworming was highest among SAC (75.6%) followed by adults (69.1%) and PSAC (51.9%) for the 9,386 (86.7%) individuals who provided the history of deworming in the previous 6 months (Table 2).

Among responders (n = 7,469) to the query of the location of deworming, 88.6% of SAC reported getting dewormed through school-based MDA, while 85.1% adults and 76.2% of PSAC were dewormed through locally purchased deworming medicines (Fig 5).

## Discussion

Our study confirms the earlier findings from Bundy et. al. [12] that an impact evaluation of MDA directed at one specific risk group, SAC, may have a significant reduction in STH prevalence across all age groups in a given community. Our study used a much larger sample size with more than 10,000 stool samples and was powered to represent the source district populations. We must emphasize though, that these observed reductions in STH prevalence/intensity, do not imply a causal relationship: that is merely inferred given the multi-year high deworming coverage, earlier estimated prevalence, and our findings.

Our analysis of pooled data from community-based surveys in 10 districts in Bangladesh found a substantial reduction in overall STH prevalence from 79.8% (2005) to 14.0% (2017–2020) across all risk groups after more than 10 years of school-based systematic biannual PCT for SAC. Despite SAC being the only targeted risk group for MDA, the data shows no statistically significant differences in STH prevalence among PSAC, SAC, and adults. Although we did not specifically explore potential impact variables on the community prevalence, we speculate that the following factors might have contributed to this observation:

1. Change in health-seeking behavior in adults, namely purchasing deworming medication for themselves and family members outside of school (correlates with data on the source of deworming for PSAC and adults). This could be attributed to the positive results of school-based MDA encouraging out-of-school villagers to seek deworming, and

2. Improved WASH factors have increased access to improved sanitation at the household level.

According to the latest WHO guidance [13], the 2030 goal for STH morbidity elimination is achieved when a country/region reaches <2% MHII. The school survey indicates that Bangladesh has achieved this goal and can halt MDA for 2 years per WHO guidance [14]. However, the population-based ICSPM data for the same risk groups in the same geographic areas

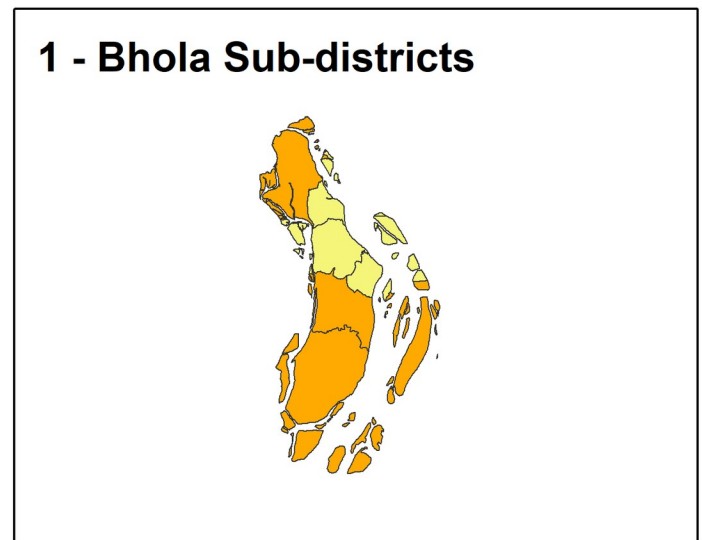

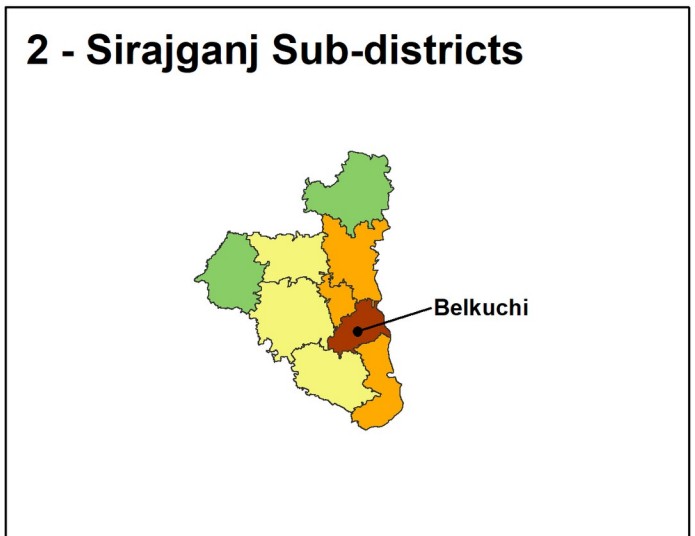

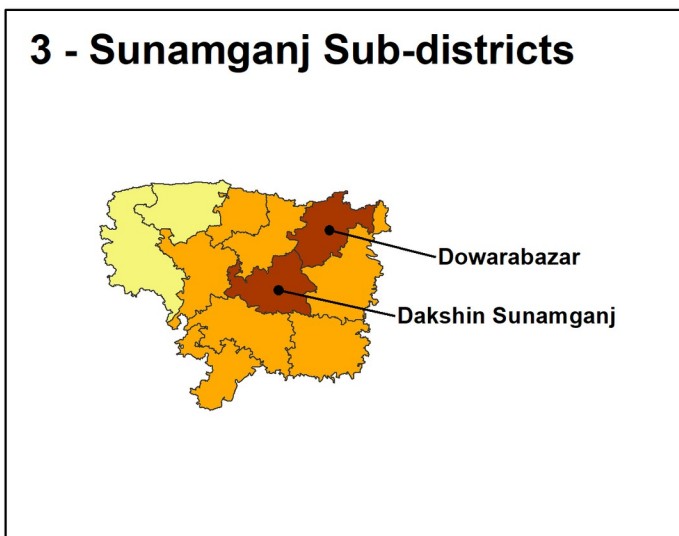

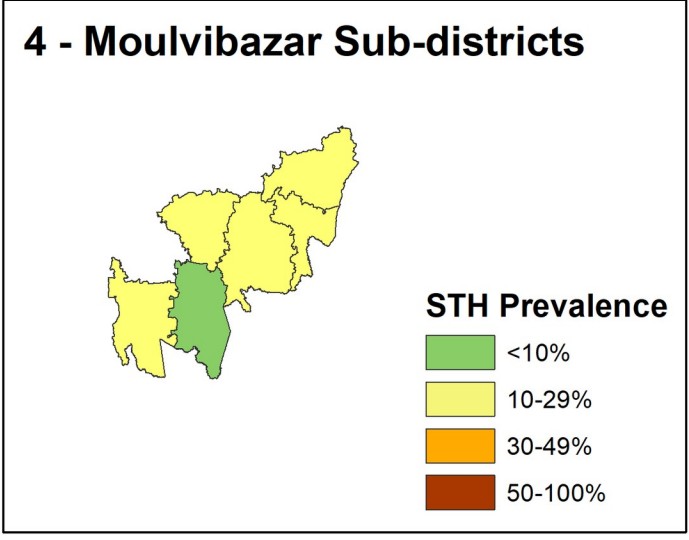

**Fig 4. STH prevalence by sub-district in the surveyed districts with the highest prevalence.**

shows the true prevalence of intensity to be still > 2. The more granular ICSPM data provides more meaningful guidance to the program, i.e. reducing MDAs in low prevalence areas but increasing interventions in clusters of high transmission that persist.

Therefore, for countries with mature programs that have reached the WHO goal of consistent coverage above 75% for 5 years, we recommend a statistically valid, population-based sampling approach to assess the sub-national level impact on prevalence and intensity of STH for use in data-driven program decision-making or policy adjustments.

**Table 2. History of deworming within the past 6 months.**

| History of deworming | Risk group N (%) | | | All Ages (%) |
|---|---|---|---|---|
|  | PSAC | SAC | Adults |  |
| Yes | 1053 (37.3) | 2733 (75.6) | 755 (25.6) | 4541 (48.4) |
| No | 1769 (62.7) | 884 (24.4) | 2192 (74.4) | 4845 (51.6) |
| **All 10-districts** | 2822 (100) | 3617 (100) | 2947 (100) | 9386 (100) |

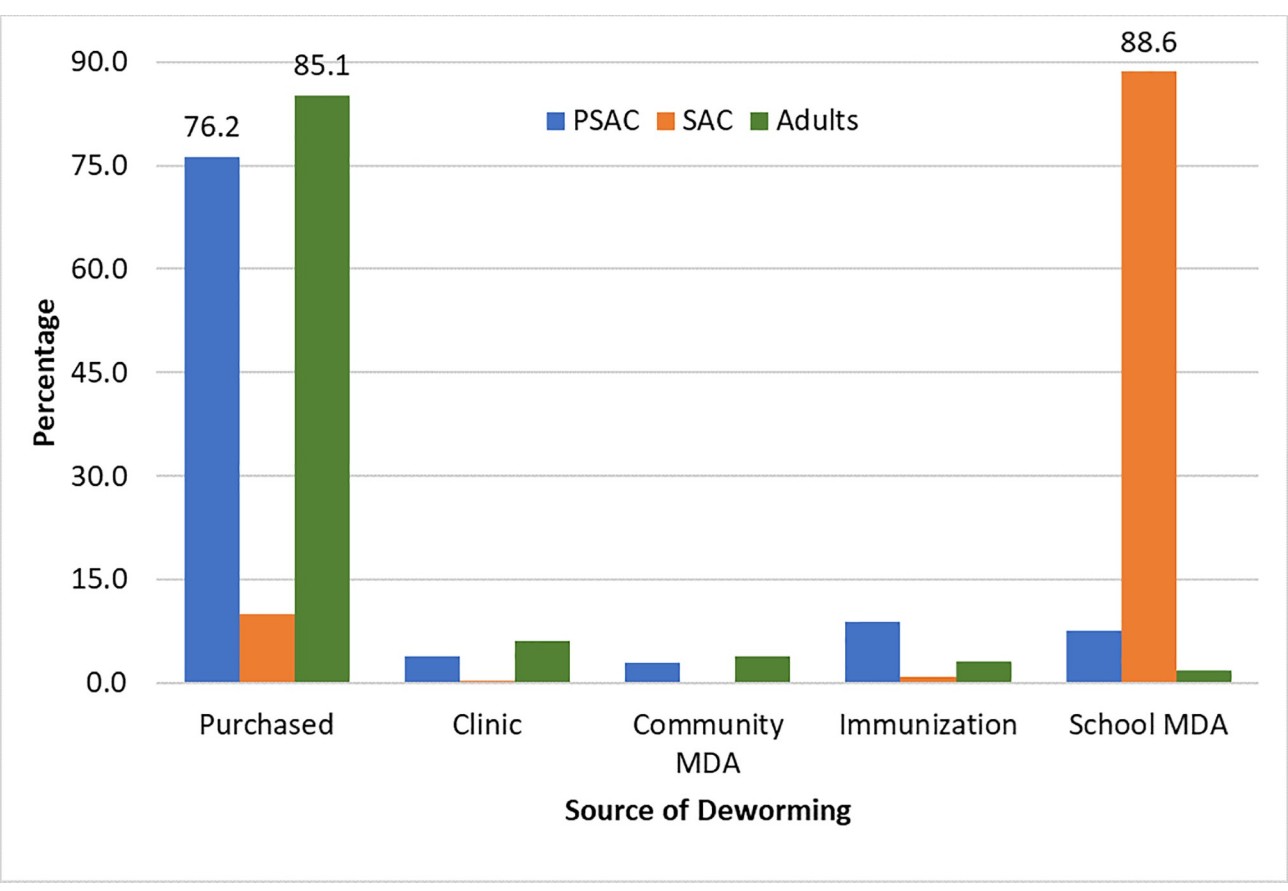

**Fig 5. Sources of deworming among those who reported receipt of deworming in the previous year.**

Additionally, our analysis revealed that the impact of STH control measures is not uniform across the country: it was significantly reduced in six districts, while the other four still carry a burden of higher prevalence and intensity. Potential factors influencing the impact of MDA on STH prevalence and intensity may be related to the local population and individual characteristics, as well as service processes related to intervention quality such as:

1. Varied baseline STH prevalence and intensity, according to a report by the Bangladesh MOHFW (2).

2. Population movement across district borders, bringing the infection from other areas.

3. The complicated relationship between drug distributors and targeted risk groups.

4. Variable environmental or ecological characteristics among districts that support longer survival of STH eggs in the soil. As an example, Sunamganj and Sirajganj districts showed persistent high STH prevalence. Both have difficult, remote terrain with poorer, less educated populations, frequent flooding, and unprotected latrines: all promoting STH re-infection and spread (Fig 6).

5. False rumors or distrust of government programs about the 'real' purpose of the treatment, and

6. Responses to local socio-cultural control measures.

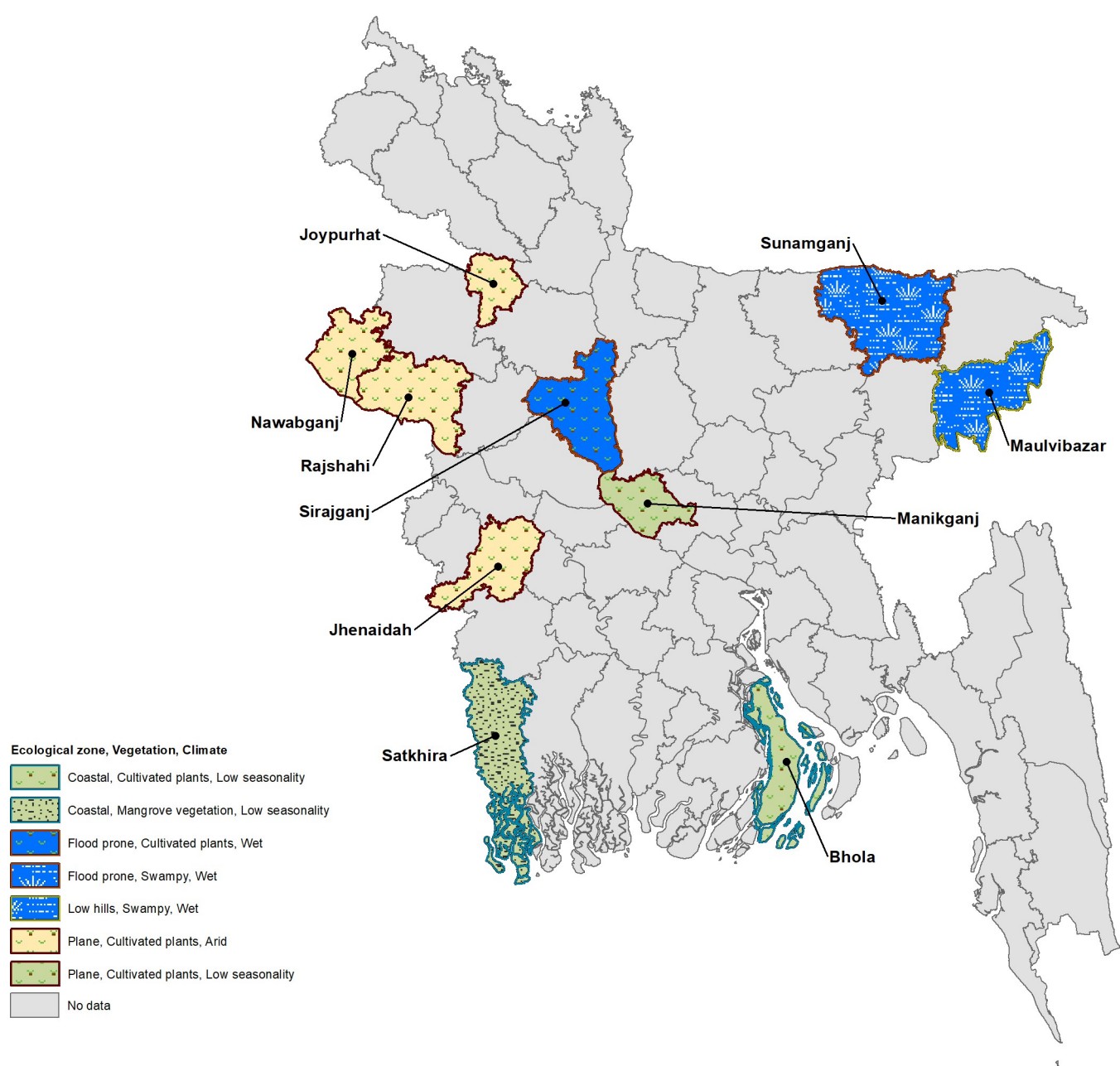

**Fig 6. Ecological zone, vegetation type, and climate in each surveyed district.**

Additional factors and changes related to differential evolution of improved sanitation, socio-economic and other development indices across and within the surveyed districts could also account for the observed differences in STH burden. While these were outside the scope of this study, we are sharing some national level developmental indices that indicate progress in key areas which could have also influenced STH outcomes. According to Joint Monitoring Program (JMP) data shared by WHO and UNICEF, Bangladesh has made significant improvements in providing access to improved latrines to 52.2% (2017) from 37.9% (2008) of the total population [15].

Similarly World Bank data shows that:

○ the Gross National Income (GNI) per capita has increased more than 165% from USD 660 (2008) to USD 1750 (2018) [16]

○ the infant mortality rate has gone down from 43.2 (2008) to 25.1 (2018) per 1,000 live births, and

○ the prevalence of undernourishment has been declining, life expectancy is continuously increasing and the literacy rate is also going up.

Among high STH prevalent districts, STH prevalence ranged from 5.1% (Kazipur sub-district, Sirajganj) to 71% (Dowara Bazar sub-district, Sunamganj). All corresponding sub-districts (Sunamganj and Bhola), seven of nine sub-districts (Sirajganj), and one of seven sub-districts (Moulvibazar) had a prevalence of more than 20%.

The Bangladesh national NTD program's STH control office plans to use these survey results to further assess the reasons for persistent high infection-transmission in some sub-districts using standard tools and mixed methods to subsequently design a focused intervention program in these locations. Additionally, the national program may use these findings to make decisions for altering the frequency of MDA programs in the low prevalence districts.

To further explore the consistent impact across all age groups, we reviewed the pill intake and their source data by age group and by district as well as collectively for all 10 districts. This revealed that a large proportion of PSAC and adults reported the purchase of locally available drugs as the primary source of deworming in the past six months. Our findings are similar to those from a school-based survey in Sri Lanka. [17] This finding has potential implications for the national program as it may indicate communal behavioral change towards self-investment in preventive health. This could be a spillover effect of the sustained impact of school deworming in these districts, signaling that school-based MDA and accompanying community messaging raises awareness of the positive health outcomes of deworming, triggering treatment-seeking behavior in community members who do not have access to school MDAs but do have local access to affordable, high quality deworming medicines (Bangladesh generic manufacturers and formulations of benzimidazoles: CWW web-survey, 2019). These initial findings need further exploration and, if confirmed, will be an important factor influencing a national policy of sustainable domestic financing guided by quality disease, socio-behavioral and pharmacological data. Similar behavioral changes should be explored by other national programs that have quality generics available locally for deworming.

## Conclusion

After 23 rounds of school-based MDA to lower the burden of STH infection since 2008, a review of survey data from 10 districts in Bangladesh shows that it is close to eliminating the infection as a public health problem from most of the country. The results of these surveys will be critical to sustain the current progress and plan corrective actions. Bangladesh plans to identify and treat all community members at risk in the persistent high-prevalence pockets of geographic areas, such as Sunamganj, Bhola, and Sirajganj. Community-based surveys may serve as better tools for advanced PCT programs to accurately assess the impact of PCT and identify hyperendemic foci that need accelerated interventions. This survey methodology provides additional valuable information on community deworming behavior which needs further validation and studies. Such population-representative results are not available from school-based survey methods.

### Limitations

The ICSPM surveys had some limitations including a higher than expected stool nonresponse rate, possible recall bias (particularly the responses to the history and location of deworming questions), and gender inequity among adult respondents. Additionally, the timing between the stool sample deposit by the survey respondents and testing in the laboratory may have been longer than ideal due to geographical challenges. This may have underestimated the hookworm prevalence slightly, but there is only one published study that documents the ideal specimen testing interval for hookworms [18] and additional studies have shown little or no hookworm in south Asia.

It is of note that Bangladesh's ELFSTH treated 19 LF-endemic districts with Albendazole (also active against STH worms) to control Lymphatic Filariasis (LF) through community-based MDAs from 2001 to 2014. While these LF-focused MDAs also impacted the STH prevalence in those 19 districts, these treatments did not affect ICSPM results as the LF program ceased in 2014 and ICSPM started data collection in 2017. Schistosomiasis is not a significant burden in Bangladesh so the ELFSTH program has not used Praziquantel in the country. Recent use of these drugs, which also affect STH, could have affected our observed results.

### Acknowledgments

We would like to express our sincere gratitude to all the staff of the respective district & sub-district health officials, field staff, and survey participants in Bangladesh, as well as the multiple CWW team members and consultants who historically contributed to protocol development and early implementation of some surveys. Technical support and field implementation was supported by CWW and the Bangladesh National ELFSTH.

### Author Contributions

**Conceptualization:** Rubina Imtiaz.

**Data curation:** Sanjaya Dhakal, Cara Tupps.

**Formal analysis:** Sanjaya Dhakal.

**Funding acquisition:** Rubina Imtiaz.

**Investigation:** Mohammad Jahirul Karim, Abdullah Al Kawsar, Ashraful Kabir, Rubina Imtiaz.

**Methodology:** Sanjaya Dhakal, Mohammad Jahirul Karim, Rubina Imtiaz.

**Project administration:** Mohammad Jahirul Karim, Abdullah Al Kawsar, Jasmine Irish, Mujibur Rahman, Ashraful Kabir, Rubina Imtiaz.

**Resources:** Jasmine Irish, Rubina Imtiaz.

**Software:** Cara Tupps.

**Supervision:** Mohammad Jahirul Karim, Abdullah Al Kawsar, Rubina Imtiaz.

**Validation:** Rubina Imtiaz.

**Visualization:** Cara Tupps.

**Writing – original draft:** Sanjaya Dhakal.

**Writing – review & editing:** Mohammad Jahirul Karim, Abdullah Al Kawsar, Jasmine Irish, Ashraful Kabir, Rubina Imtiaz.

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
