## [Decision Letter · Decision Letter 0]

18 Aug 2020

Dear Dr. Imtiaz,

Thank you very much for submitting your manuscript "Post-intervention Epidemiology of STH in Bangladesh: data to sustain the gains" for consideration at PLOS Neglected Tropical Diseases. As with all papers reviewed by the journal, your manuscript was reviewed by members of the editorial board and by several independent reviewers. The reviewers appreciated the attention to an important topic. Based on the reviews, we are likely to accept this manuscript for publication, providing that you modify the manuscript according to the review recommendations. 

Sincerely,

Antonio Montresor

Associate Editor

Marco Coral-Almeida

Deputy Editor

Reviewer's Responses to Questions

**Key Review Criteria Required for Acceptance?**

**Methods**

-Are the objectives of the study clearly articulated with a clear testable hypothesis stated?

-Is the study design appropriate to address the stated objectives?

-Is the population clearly described and appropriate for the hypothesis being tested?

-Is the sample size sufficient to ensure adequate power to address the hypothesis being tested?

-Were correct statistical analysis used to support conclusions?

-Are there concerns about ethical or regulatory requirements being met?

Reviewer #1: See below

Reviewer #2: In various places (eg pages 9 and 12) it is stated that 12 Districts were surveyed and data from 10 of these are reported here. I can't see the criteria for reporting a subset of data. If this is a partial view based on a decision of the authors this could result in significant bias. A clear explanation is essential.

It would be helpful to cite the sources of ethical clearance in the methods (they are attached to the file, but should be transparent to the reader of the paper too).

Reviewer #3: (No Response)

**Results**

-Does the analysis presented match the analysis plan?

-Are the results clearly and completely presented?

-Are the figures (Tables, Images) of sufficient quality for clarity?

Reviewer #1: See below

Reviewer #2: P13 line 80 states "According to the 2011 national population census, this analysis

represents about 15.5% of the Bangladesh population". It is unclear what this means. The sample is around 11,000 people, the population of Bangladesh is 160 million, so the sample is much less than 1%. 

The enrolled sample is 17k and the specimens submitted are 11k, so almost 40% unsampled. Collecting stool specimens is notoriously difficult, but this seems a particularly low compliance. It is necessary to present an analysis of whether the compliance conceals bias: eg by gender, by age, locality etc. So that an assessment can be made of the consequences of the 40% under-sampling.

Reviewer #3: (No Response)

**Conclusions**

-Are the conclusions supported by the data presented?

-Are the limitations of analysis clearly described?

-Do the authors discuss how these data can be helpful to advance our understanding of the topic under study?

-Is public health relevance addressed?

Reviewer #1: See below

Reviewer #2: page 17 line 130 states:To our knowledge, this is the first time an impact evaluation of MDA directed at one

specific risk group, SAC, has shown a significant reduction in STH prevalence across all age groups in a given community. Suggest checking this reference which shows this: Bundy, D.A.P., Wong, M.S., Lewis, L. and Horton, J. (1990) Control of geohelminths by delivery of targeted chemotherapy through schools. Transactions of the Royal Society of Tropical Medicine and Hygiene 84, 115-120.

page 19 line 176: This is a rather limited list of possible explanations for variatiosn in outcomes across districts. I am particularly surprised to see no mention of two factors known to be important: WASH coverage and economics. At present the discussion is speculative and uninformative. The authors should make this point quantitatively. Data are available for these 10 districts which would allow comparison/ranking using government and/or world bank data on both WASH and the economy. 

page 20 194-196: the point about the transition from MDA to a mixture of self-purchasing in better off areas and MDA in poorer is important. There should be specific reference here to the work of Nilanthi Da Silva on exactly this topic in Sri Lanka, where public MDA only now persists in the poorest "plantation" areas. If the authors here rank districts by economics (the point above) then they could make this call specifically.

Reviewer #3: (No Response)

**Editorial and Data Presentation Modifications?**

Reviewer #1: See below

Reviewer #2: NA

Reviewer #3: (No Response)

**Summary and General Comments**

Reviewer #1: General: Dhakal and co-authors present a well-written account of a rigorous survey on soil-transmitted helminth (STH) prevalence and selected risk factors across Bangladesh. Such high-quality studies are invaluable for program management, and are examples for other programs facing the question whether to continue interventions or adapt to changing conditions. 

A few comments are offered for clarification and discussion:

- Line 45: It is unclear how the survey districts were selected

- Lines 50-51: supposedly, the word “years old” is missing after the figure

- Line 70: provide an estimate on the time lapse between sample production and sample analysis (e.g. “analyzed on the day of sample collection”)

- Line 85: was the survey really powered to detect a certain prevalence of MHII?

- Line 94: the figure given in the abstract and here on the number of individuals submitting stool samples is not identical

- Line 99: basic data on the ~40% who did not provide samples should be provided (variation between districts, age class and sex distribution)

- Line 104: Ascaris, Trichuris: please write the proper species name/abbreviation

- Line 131: the data are intriguing but there is no evidence for causality between school-based MDA and the prevalence reduction. Some points are discussed below, and the authors should be careful to not over-interpret their data or suggest causality. 

- Are there any plans to sample districts that were not covered in this survey?

- Is there any information on the timing of sample collection compared to the last deworming event?

- Some abbreviations are used without introduction, or they are introduced after their first use (e.g. ICSPM, ELFSTH)

Reviewer #2: This paper explores an important and topical issue. The finding that treating school children only has an impact on transmission as a whole was made some 40 years ago by R.M.Anderson, but has been demonstrated only rarely (but it has been demonstrated before - see reference in comments above). There is a current trend in the literature to argue for treatment of all ages as essential to significant interruption of transmission to significantly lower levels. This paper is one of a very few to show that this is not necessary. This is such an important and topical point that I am surprised that the authors do not make more of it. School based treatment is so much less costly and easier, which is why the programme has been self-sustainable in Bangladesh.

On a related point, a very novel finding of this study is the very high rates of adult and PSAC self-treatment using out-of-pocket funds. This supports the contention that school based public treatment will be supported by out-of-pocket self treatment by those who can afford to do so. This was the experience in Sri Lanka that led to their being able to refine, reduce and target their school based MDA programme. These are important policy implications that deserve stronger mention.

In describing the outcomes the authors use the figure of 79.8% as the baseline figure from 2005, based on a government report. The details of this baseline survey need to be described here. Was this for the same 10 Districts as here; was it the same age groups; same Dx? etc The authors need to be able to make the case that the comparison with the present result is valid. If it is not, then the very low prevalence seen today still stands as a cross country sampling of the status of infection after treatment, but clearly needs to be presented in that way.

Reviewer #3: Comments and observations

Overall, the manuscript is relevant, well written and with study design according to WHO recommendations. It provides useful information regarding the impact of more than 10 years mass drug administration interesting on prevalence and intensity of STH infections in Bangladesh. Nevertheless, the manuscript needs to be improved regarding some aspects of methodology, results, and discussion.

Methodology: 

1) It should be important to describe whether the different districts and communities have comparable characteristics or are in different settings: ecological settings or variation, vegetation, climate, seasonality and water contact activities and behavior. 

2) Describe the main occupation of the population and their likely exposure to STH transmission. 

3) If possible, the authors should give some information on health facilities in relation to STH: diagnostic facilities, availability of praziquantel. 

4) Provide short information regarding WASH activities, hygiene, and sanitation. 

5) It is not clear whether the treatment has been provided to the children following the examination

Results

6) The authors should provide additional Table(s) showing the results of prevalence and if possible, the intensity of STH infections in 2005 compared to 2017-2020 in different district to justify the objective of the study.

Discussion

7) Results of differences in prevalence and intensity need to be more explained. 

• Why the highest STH prevalence in Sunamganj, Bhola and Sirjganj in contrast to others?

• The same question goes for the intensity.

8) What do specifically, the authors mean by ecological variation? 

Recommendation

The manuscript could be accepted after considering the comments and observations with “major changes”

PLOS authors have the option to publish the peer review history of their article (what does this mean?). If published, this will include your full peer review and any attached files.

Reviewer #1: Yes: Peter Steinmann

Reviewer #2: No

Reviewer #3: Yes: Moussa Sacko, PhD
---

## [Decision Letter · Decision Letter 1]

23 Oct 2020

Dear Dr. Imtiaz,

We are pleased to inform you that your manuscript 'Post-intervention epidemiology of STH in Bangladesh: data to sustain the gains' has been provisionally accepted for publication in PLOS Neglected Tropical Diseases.

Best regards,

Antonio Montresor

Associate Editor

Marco Coral-Almeida

Deputy Editor

Reviewer's Responses to Questions

**Key Review Criteria Required for Acceptance?**

**Methods**

-Are the objectives of the study clearly articulated with a clear testable hypothesis stated?

-Is the study design appropriate to address the stated objectives?

-Is the population clearly described and appropriate for the hypothesis being tested?

-Is the sample size sufficient to ensure adequate power to address the hypothesis being tested?

-Were correct statistical analysis used to support conclusions?

-Are there concerns about ethical or regulatory requirements being met?

Reviewer #1: See below

Reviewer #2: (No Response)

Reviewer #3: (No Response)

**Results**

-Does the analysis presented match the analysis plan?

-Are the results clearly and completely presented?

-Are the figures (Tables, Images) of sufficient quality for clarity?

Reviewer #1: See below

Reviewer #2: (No Response)

Reviewer #3: (No Response)

**Conclusions**

-Are the conclusions supported by the data presented?

-Are the limitations of analysis clearly described?

-Do the authors discuss how these data can be helpful to advance our understanding of the topic under study?

-Is public health relevance addressed?

Reviewer #1: See below

Reviewer #2: (No Response)

Reviewer #3: (No Response)

**Editorial and Data Presentation Modifications?**

Reviewer #1: See below

Reviewer #2: (No Response)

Reviewer #3: (No Response)

**Summary and General Comments**

Reviewer #1: The authors have adequately addressed the comments by this reviewer.

Reviewer #2: The authors have responded effectively to the reviewers' comments.

Reviewer #3: This is an already revised version of the manuscript. The manuscript is relevant, well written and with study design according to WHO recommendations. It provides useful information regarding the impact of more than 10 years mass drug administration interesting on prevalence and intensity of STH infections in Bangladesh. Overall, the authors have addressed all the points raised and the explanation are acknowledged.

I am sorry for the question No3 regarding praziquantel in methodology. It was related to the availability of drugs recommended for STH (i.e. albendazole or mebendazole) and not praziquantel.

PLOS authors have the option to publish the peer review history of their article (what does this mean?). If published, this will include your full peer review and any attached files.

Reviewer #1: **Yes: **Peter Steinmann

Reviewer #2: No

Reviewer #3: No

---

## [Editor Report · Acceptance letter]

24 Nov 2020

Dear Dr. Imtiaz,

We are delighted to inform you that your manuscript, "Post-intervention epidemiology of STH in Bangladesh: data to sustain the gains," has been formally accepted for publication in PLOS Neglected Tropical Diseases.

Best regards,

Shaden Kamhawi

co-Editor-in-Chief

Paul Brindley

co-Editor-in-Chief
